# RecursiveMix: Mixed Learning with History

**Lingfeng Yang**[1#], **Xiang Li**[2,1#], **Borui Zhao**[3], **Renjie Song**[3], **Jian Yang**[1*]

[1]Nanjing University of Science and Technology, [2]Nankai University, [3]Megvii Technology

{yanglfnjust, csjyang}@njust.edu.cn, xiang.li.implus@nankai.edu.cn
zhaoborui.gm@gmail.com, songrenjie@megvii.com

## Abstract

Mix-based augmentation has been proven fundamental to the generalization of deep vision models. However, current augmentations only mix samples from the current data batch during training, which ignores the possible knowledge accumulated in the learning history. In this paper, we propose a recursive mixed-sample learning paradigm, termed "RecursiveMix" (RM), by exploring a novel training strategy that leverages the historical input-prediction-label triplets. More specifically, we iteratively resize the input image batch from the previous iteration and paste it into the current batch while their labels are fused proportionally to the area of the operated patches. Furthermore, a consistency loss is introduced to align the identical image semantics across the iterations, which helps the learning of scale-invariant feature representations. Based on ResNet-50, RM largely improves classification accuracy by $\sim$3.2% on CIFAR-100 and $\sim$2.8% on ImageNet with negligible extra computation/storage costs. In the downstream object detection task, the RM-pretrained model outperforms the baseline by 2.1 AP points and surpasses CutMix by 1.4 AP points under the ATSS detector on COCO. In semantic segmentation, RM also surpasses the baseline and CutMix by 1.9 and 1.1 mIoU points under UperNet on ADE20K, respectively. Codes and pretrained models are available at https://github.com/implus/RecursiveMix.

## 1 Introduction

Deep convolutional neural networks (CNN) have made great progress in many computer vision tasks such as image classification [26, 29, 62], object detection [25, 71, 42], semantic segmentation [72, 67], etc. Despite achieving better performances, the models inevitably become more complex and incur over-fitting risks. To overcome this problem, methods like regularization [53, 30] and data augmentation [15, 70, 68] are developed consequently. Recently, mixed-based data augmentations [70, 68, 59, 60, 34, 13] have been proven to enhance the model generalization capability. Following the popular Mixup [70] and CutMix [68], which optimize networks based on mixed pixels and fused labels of two images, improvement has been made on aspects of the mix level [59, 18, 39] and mix region [34, 13, 60]. However, existing methods only mix samples from the current data batch. As a result, they fail to capitalize on the potential knowledge in their learning history.

To leverage the historical knowledge, we propose an efficient and effective augmentation approach termed "RecursiveMix" (RM) that makes use of the historical *input-prediction-label* triplets. Different

---

*Corresponding author. # Equal contributions. Lingfeng Yang, Xiang Li and Jian Yang are with Jiangsu Key Lab of Image and Video Understanding for Social Security, and Key Lab of Intelligent Perception and Systems for High-Dimensional Information of Ministry of Education, Nanjing University of Science and Technology, Nanjing, 210094, P.R. China.

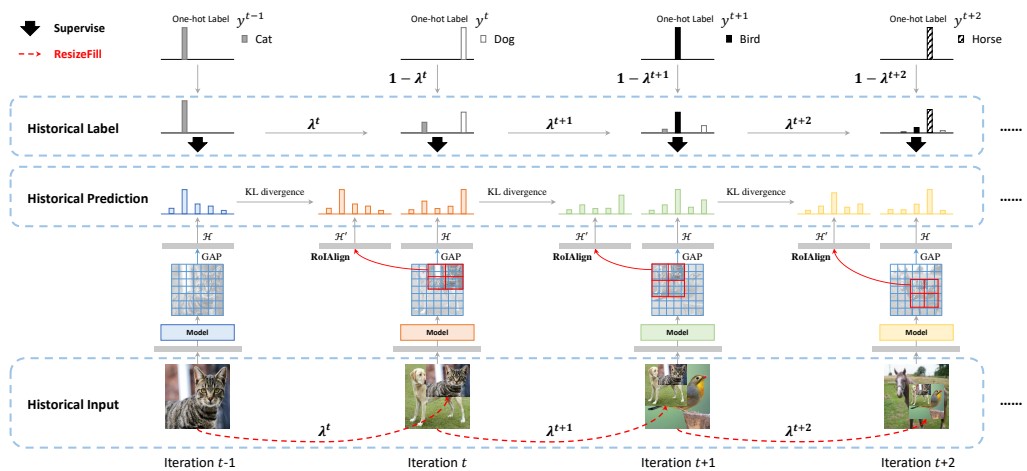

Figure 1: Illustration of the RecursiveMix, which leverages the historical input-prediction-label triplets in a recursive paradigm. The historical input images are resized and then mixed into current ones where labels are fused proportionally ($\lambda$ and $1 - \lambda$) to the area of operating patches.

from the conventional mix-based approaches which only fuse the training images in the current batch, RM further leverages the last mixed inputs and labels to form new ones in a recursive paradigm. It aims at continuously reusing the augmented data and supervisions in previous iterations (Fig. 1). In addition, a consistency loss is introduced to align two spatial representations derived from the identical instance. RM shows significant advantages over its competitive counterparts [68, 70] on classification tasks and its space-aware features can potentially benefit the downstream tasks such as object detection and segmentation as they are more sensitive to the spatial understanding of the image. The advanced performance of RM will be illustrated later in our methods and experiments.

Another notable property of the proposed RM is that it hardly increases the training/inference budget (Table 11) but only consumes an extremely negligible amount of storage complexity (only a mini-batch of the historical input, prediction, and label) and a lightweight additional module (only an RoI [25] operator and an optional fully connected layer). Experimental evidence in CIFAR and ImageNet classification benchmarks demonstrate its effectiveness. Specifically, RM shows a gain of absolutely 3.2% points over the baseline model and outperforms CutMix [68] by 1.1% points in the CIFAR-100 dataset under DenseNet-161 [29]. In the ImageNet dataset, RM improves by 2.9% and 2.7% points based on the ResNet-50 [26] and PVTv2-B1 [61] baselines while outperforming CutMix by 0.6% and 0.7% points, respectively. In terms of the downstream tasks, RM-pretrained models improve the baseline by 1∼3 box AP points (object detection) and 1∼2 mask AP points (instance segmentation) on the COCO dataset and 1∼3 mIoU (semantic segmentation) on ADE20K while consistently outperforming models pretrained by CutMix. Notably, under the one-stage detector ATSS [71], the RM-pretrained model under ResNet-50 outperforms the baseline and CutMix by 2.1 AP and 1.4 AP points, respectively.

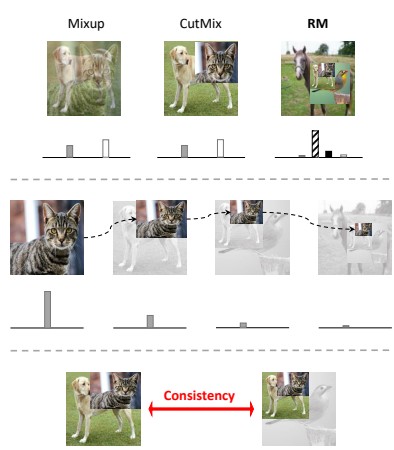

Figure 2: Three benefits of the proposed RecursiveMix, which iteratively leverages the historical input-prediction-label triplets: 1) an enlarged diversity of data space and richer supervisions, 2) adequate training signals for an instance with multi-scale-/-space views, and 3) explicit learning on the spatial semantic consistency which can further benefit the downstream tasks.

## 2   Related Work

**Regularization Methods.** There are many other types of regularization techniques that focus on the feature [53, 20, 30], data [15, 8], label [49], data-label pair [70, 68, 34] and feature-label pair [59]. Dropout [53], DropBlock [20] and Stochastic Depth [30] introduce the stochastic feature drop during training in an element-wise, block-wise, and path-wise way, respectively. Cutout [15] and GridMask [8] randomly/systematically erase regional pixels on images while Label Smoothing [49]

tries to prevent the classifiers from being too confident on a certain category by slightly modifying the training labels into a soft distribution [27]. Mixup [70] and CutMix [68] both combine two samples, where the corresponding training label is given by the linear interpolation of two one-hot labels. They differ in the detailed combining strategy of the input images: Mixup employs pairwise linear combination but CutMix adopts a regional crop-and-paste operation. SaliencyMix [58], PuzzleMix [34], Co-Mixup [33], and SuperMix [13] further exploit the salient regions and local statistics for optimal Mixup. Attentive CutMix [60] and SnapMix [31] take the activation map of the correct label as a confidence indicator for selecting the semantically meaningful mix regions. TransMix [5] requires the attention map of the self-attention module to re-weight the targets. Manifold Mixup [59], PatchUp [18] and MoEx [39] perform a feature-level interpolation over two samples to prevent overfitting the intermediate representations. Recently, StyleMix [28] employs style transfer to enrich the generated mixed images. The proposed RecursiveMix (RM) shares similarities with CutMix by combining data samples in a regional replacement manner, but differs essentially from all the existing regularizers in that RM is the first to exploit the historical information. In the experiment part, we will show the comparisons between the proposed RM and other regularization methods.

**Contrastive Learning.** Contrastive learning [22] aims to minimize the distances between the positive pairs by consistency losses [22] which measure their similarities in a representation space or a probability distribution space. In recent years, contrastive learning has made a lot of progress in the fields of computer vision [24, 51, 2, 57, 1, 9, 66, 50, 4, 10, 19, 11, 21, 38] and natural language processing [64, 48, 32]. Positive pairs can be constructed in many ways. In the fields of self-supervised learning and semi-supervised learning, many works [2, 24, 9, 4, 21] have used two augmented inputs of one instance to form positive pairs. In terms of generating positive pairs with different model structures or weights, [64, 52] employ Dropout [53] twice on one model, and [24, 57] make a difference in weights through Exponential Moving Average (EMA) and separate projection heads. The above methods essentially require Siamese networks [3] or the need to pass one instance through a model twice to obtain positive pairs, which involves additional computational costs and large architectural designs. Another way is to record historical representations or predictions and compare them with their current outputs [66, 35, 38, 65]. However, using a memory bank to store key statistics for all instances consumes a huge memory cost and is not applicable to even bigger datasets. Our method employs the idea of contrastive learning from the learning history while consuming extremely negligible storage and computational complexity.

**Learning with History.** In the training history, the accumulated elements are a large amount of wealth for the improvement of learning deep models. Existing methods have explored the usage across model parameters [4, 10, 24, 19, 11, 21], gradients [54, 17, 69, 36, 47, 44], predictions [35, 38], and all-level feature maps [63, 65, 66, 24] applied during the training process. More related works refer to the Supplementary Material. Different from existing practices, we exploit a novel aspect of the historical input-prediction-label triplets to improve the generalization of vision models.

## 3 Method

In this section, we introduce the proposed RecursiveMix (RM) approach and discuss its properties.

### 3.1 RecursiveMix

We attempt to exploit the historical input-prediction-label triplets of a learning instance to improve the generalization of vision models. A practical idea is to reuse the past input-prediction-label triplets to construct diversely mixed training samples with the current batch (see Fig. 1). To be specific, the historical input images are resized and then filled into the current ones where the labels are fused proportionally to the area of the mixed patches. Interestingly, such an operation formulates an exact recursive paradigm where each instance can have a gradually shrunken size of view during training, thus we term the method "RecursiveMix" (RM). Specifically at iteration $t$, the preliminary of RM is to generate a new

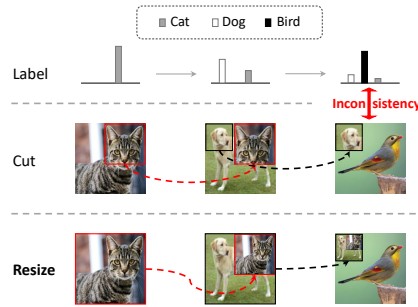

Figure 3: "Cut" operation may lead to the inconsistency between input and label under the proposed historical operation, but "Resize" can correctly preserve the consistency.

training sample $(\widetilde{x}^t, \widetilde{y}^t)$ by combining the current sample $(x^t, y^t)$ and the past historical one $(x^h, y^h)$. The new training sample $(\widetilde{x}^t, \widetilde{y}^t)$ participates in training with the original loss objectives $\mathcal{L}_{CE}$ and then updates the historical pair $(x^h, y^h)$ iteratively:

$$
\begin{aligned}
\widetilde{x}^t &= \mathbf{M} \odot \text{ResizeFill}(x^h, \mathbf{M}) + (\mathbf{1} - \mathbf{M}) \odot x^t, \\
\widetilde{y}^t &= \lambda^t y^h + (1 - \lambda^t) y^t, \\
x^h &= \widetilde{x}^t, \\
y^h &= \widetilde{y}^t,
\end{aligned}
\tag{1}
$$

where $\mathbf{M} \in \{0, 1\}^{W \times H}$ is the binary mask indicating the place for the historical input to fill, $\mathbf{1}$ is the binary mask filled with ones and $\odot$ is the element-wise product. The combination ratio $\lambda$ ($\lambda^t$ indicates $\lambda$ at moment $t$) is sampled from the uniform distribution $U[0, \alpha]$ where $\alpha$ defaults to 0.5. This setting is different from the symmetric beta distribution in Mixup [70] and CutMix [68] because $\lambda$ is not suggested to be quite large for it denotes the proportion of the historical images in our case. Otherwise, the information of the new data points has a risk of being discarded. The function $\text{ResizeFill}(x^h, \mathbf{M})$ means resizing the image $x^h$ and filling it back exactly into its rectangle sub-region where $\mathbf{M} = 1$. The resized height and width are also determined by the rectangle area. This design is to make the recursive mechanism reasonable since the original cut operation in CutMix [68] could possibly lose former information, which would cause the inconsistency problem with the label (see Fig. 3).

The binary mask $\mathbf{M}$ is generated by filling in with 1 inside the sampled bounding box coordinates $\mathbf{B} = (r_x, r_y, r_w, r_h)$, otherwise defaulting to 0. Similar to CutMix [68], the size and position of the rectangular region $\mathbf{B}$ are uniformly sampled according to:

$$
\begin{aligned}
r_x &\sim U(0, W), r_w = W\sqrt{\lambda}, \\
r_y &\sim U(0, H), r_h = H\sqrt{\lambda}.
\end{aligned}
\tag{2}
$$

As depicted in Fig. 2, the resize-filled part within the current input is semantically consistent with the historical input regardless of proportion and scale. Inspired by contrastive learning [22, 24, 4], we optimize the KL divergence between the cross-iteration predictions of the two corresponding regions. To be specific, we record the historical prediction $p^h$ which is updated by the original outputs after network function $\mathcal{F}(\cdot)$, global average pooling $\text{GAP}(\cdot)$, and a linear layer $\mathcal{H}(\cdot)$ in the last iteration:

$$
\begin{aligned}
\widetilde{p}^{t-1} &= \mathcal{H}(\text{GAP}(\mathcal{F}(\widetilde{x}^{t-1}))), \\
p^h &= \widetilde{p}^{t-1}.
\end{aligned}
\tag{3}
$$

In the current iteration, we obtain the corresponding prediction of local features by 1×1 RoIAlign [25] aligning with the computed coordinates $\mathbf{B}$ of RecursiveMix:

$$
\widetilde{p}^t_{roi} = \mathcal{H}'(\text{RoIAlign}(\mathcal{F}(\widetilde{x}^t), \mathbf{B})),
\tag{4}
$$

where $\mathcal{F}$ denotes the backbone function and $\mathcal{H}, \mathcal{H}'$ denote the final linear classification layer. By default, the parameters are not shared between layers $\mathcal{H}$ and $\mathcal{H}'$, which is purely suggested by the experiments in Table 4a. The total loss function to train our model is:

$$
\mathcal{L} = \mathcal{L}_{CE}(\widetilde{x}^t, \widetilde{y}^t) + \omega \lambda^t \mathcal{L}_{KL}(\widetilde{p}^t_{roi}, p^h),
\tag{5}
$$

where $\mathcal{L}_{CE}$ denotes the cross-entropy loss and $\mathcal{L}_{KL}$ denotes the consistency loss. Note that $p^h$ is the historical prediction from the last iteration, and the consistency loss weight $\omega$ is set to 0.1 unless otherwise stated. In addition, we also use the mixed ratio $\lambda$ representing the proportion of the resize-filled historical image to weight $\mathcal{L}_{KL}$. It makes sense that the confidence of $\mathcal{L}_{KL}$ relates to $\lambda$ as a smaller $\lambda$ (smaller image size to be filled) usually causes the RoI feature to lose more spatial semantic information.

## 3.2 Discussion

Similar to CutMix/Mixup, RM is simple and hardly introduces additional computational costs (see Table 11). During training, only a mini-batch of the historical input, label, and prediction (i.e., $(x^h, y^h, p^h)$) needs to be stored and updated iteratively. The RoI [25] operator and an individual fully connected layer are considerably lightweight and free of deployment. The additional storage cost is extremely negligible compared to the entire model, data, and feature maps, thus making RM very efficient to train any network architecture.

As illustrated in Fig. 2, introducing the historical design of input, label, and prediction into augmenting the mixed training pairs recursively and bridging the spatial semantics consistency has three obvious

| PyramidNet-200, $\tilde{\alpha}$=240 (300 epochs) | Top-1 Err (%) |
|---|---|
| Baseline | 3.85 |
| + Label Smoothing [49] | 3.74 |
| + DropBlock [20] | 3.27 |
| + Stochastic Depth [30] | 3.11 |
| + Cutout [15] | 3.10 |
| + Mixup ($\alpha$=1.0) [70] | 3.09 |
| + Manifold Mixup ($\alpha$=1.0) [59] | 3.15 |
| + CutMix [68] | 2.88 |
| + MoEx [39] | 3.44 |
| + StyleCutMix (auto-$\gamma$) [28] | 2.55 |
| + RM (ours) | **2.35** |

Table 1: Comparison of state-of-the-art regularization methods in CIFAR-10 under 300 epochs settings, the same as that of the Cut-Mix [68] paper. Results are reported as average over 3 runs.

| Model | RS | HIS | CL | Top-1 Err (%) |
|---|---|---|---|---|
| PyramidNet | – | – | – | 16.67 |
| + CutMix [68] | | | | 15.59 |
| + RM (ours) | ✓ | | | 15.36 |
| | ✓ | ✓ | | 14.81 |
| | ✓ | ✓ | ✓ | **14.65** |

Table 2: Ablation studies on the individual gain by each component of RM in CIFAR-100 with PyramidNet-164, $\tilde{\alpha}$=270 under the 200-epoch setting. The results are averaged over 3 runs. "**RS**": Resize strategy. "**HIS**": Historical mix. "**CL**": Consistency loss.

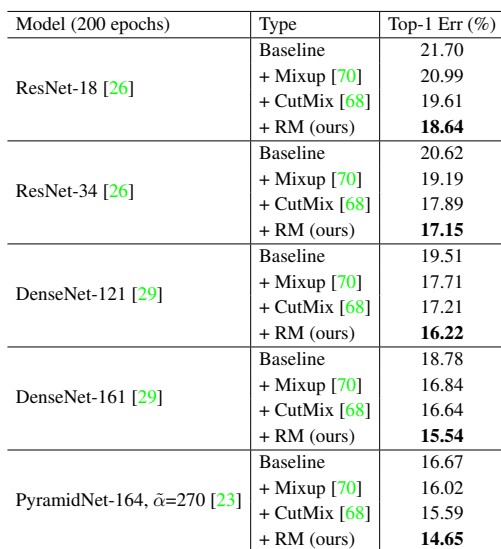

| Model (200 epochs) | Type | Top-1 Err (%) |
|---|---|---|
| ResNet-18 [26] | Baseline | 21.70 |
| | + Mixup [70] | 20.99 |
| | + CutMix [68] | 19.61 |
| | + RM (ours) | **18.64** |
| ResNet-34 [26] | Baseline | 20.62 |
| | + Mixup [70] | 19.19 |
| | + CutMix [68] | 17.89 |
| | + RM (ours) | **17.15** |
| DenseNet-121 [29] | Baseline | 19.51 |
| | + Mixup [70] | 17.71 |
| | + CutMix [68] | 17.21 |
| | + RM (ours) | **16.22** |
| DenseNet-161 [29] | Baseline | 18.78 |
| | + Mixup [70] | 16.84 |
| | + CutMix [68] | 16.64 |
| | + RM (ours) | **15.54** |
| PyramidNet-164, $\tilde{\alpha}$=270 [23] | Baseline | 16.67 |
| | + Mixup [70] | 16.02 |
| | + CutMix [68] | 15.59 |
| | + RM (ours) | **14.65** |

Table 3: Performance on various architectures in CIFAR-100 under 200 epochs. The proposed RM ($\alpha$=0.5, $\omega$=0.1) shows consistent improvements over other competitive approaches. We report an average of 3 runs.

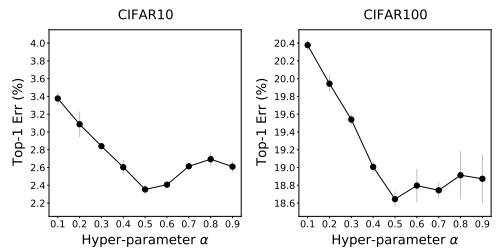

Figure 4: The effect of hyperparameter $\alpha$ on CIFAR-10 and CIFAR-100 datasets. Standard deviation is also plotted and another hyperparameter $\omega$ is fixed to 0.1.

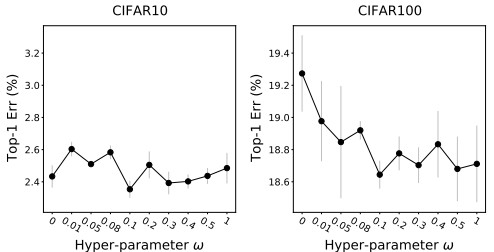

Figure 5: The effect of hyperparameter $\omega$ on CIFAR-10 and CIFAR-100 datasets. Standard deviation is also plotted and another hyperparameter $\alpha$ is fixed to 0.5.

advantages: 1) RM encourages more diversity in data space as it exhaustively explores exponential combinations of multiple instances along with their corresponding richer training signals within a single iteration. 2) RM can explicitly provide continuous training supervisions for an instance with multi-scale and spatial-variant views. 3) RM makes use of the identical part of inputs from two adjacent iterations through contrastive learning via the consistency loss, which helps the model learn spatial-correlative semantic representation. These properties thus lead to the better generalization of computer vision models and can potentially benefit downstream tasks, e.g., object detection and semantic segmentation with enhanced spatial representation ability. Interestingly these properties can be simultaneously achieved by a simple operation of iteratively using the historical input-prediction-label triplets.

# 4 Experiment

In this section, we evaluate RM on image recognition tasks to show the effectiveness of the usage of historical input-prediction-label triplets. More details can be found in the Supplementary Material.

## 4.1 CIFAR Classification

**Dataset.** The two CIFAR datasets [37] consist of colored natural scene images, each with 32×32 pixels in total. The train and test sets have 50K images and 10K images respectively. CIFAR-10 has 10 classes and CIFAR-100 has 100.

**Setup.** We conduct two major training settings: 200-epoch and 300-epoch training, respectively. For 200-epoch training, we employ SGD with a momentum of 0.9, a weight decay of $5 \times 10^{-4}$, and

| Top-1 Err (%) | CIFAR-10 | CIFAR-100 | ImageNet |
|---|---|---|---|
| Unshared weights | **2.35** | **18.64** | **20.80** |
| Shared weights | 2.61 | 18.66 | 20.92 |

(a) Comparisons of linear layer weights.

| Top-1 Err (%) | CIFAR-10 | CIFAR-100 | ImageNet |
|---|---|---|---|
| nearest | **2.35** | **18.64** | **20.80** |
| bilinear | 2.58 | 18.78 | 20.86 |

(b) Comparisons of different interpolation modes.

Table 4: Ablation studies on 1) CIFAR-10 with PyramidNet-200, $\tilde{\alpha}$=240 under 300-epoch training setting, 2) CIFAR-100 with ResNet-18 under 200-epoch training setting and 3) ImageNet with ResNet-50 under 300-epoch training setting.

| Model (300 epochs) | Top-1 Err (%) | Top-5 Err (%) |
|---|---|---|
| ResNet-50 [70] | 23.68 | 7.05 |
| + Mixup [70] | 22.58 | 6.40 |
| + CutMix [68] | 21.40 | 5.92 |
| + RM (ours) | **20.80** | **5.42** |
| PVTv2-B1 [61] | 24.92 | 8.06 |
| + Mixup [70] | 23.23 | 6.65 |
| + CutMix [68] | 22.92 | 6.42 |
| + RM (ours) | **22.22** | **6.17** |

Table 5: Performance on various architectures in ImageNet under 300 epochs. The proposed RM ($\alpha$=0.5, $\omega$=0.5) shows consistent improvements over other competitive counterparts.

| ResNet-50 (300 epochs) | Top-1 Err (%) | Top-5 Err (%) |
|---|---|---|
| Baseline | 23.68 | 7.05 |
| + Cutout [15] | 22.93 | 6.66 |
| + Stochastic Depth [30] | 22.46 | 6.27 |
| + Mixup [70] | 22.58 | 6.40 |
| + Manifold Mixup [59] | 22.50 | 6.21 |
| + DropBlock [20] | 21.87 | 5.98 |
| + Feature CutMix [68] | 21.80 | 6.06 |
| + CutMix [68] | 21.40 | 5.92 |
| + PuzzleMix [34] | 21.24 | 5.71 |
| + MoEx [39] | 21.90 | 6.10 |
| + CutMix + MoEx [39] | 20.90 | 5.70 |
| + RM (ours) | **20.80** | **5.42** |

Table 6: Comparison of state-of-the-art regularization methods in ImageNet under 300 epochs.

2 GPUs with a mini-batch size of 64 on each to optimize the models. The learning rate is set to 0.1 with a linear warmup [26] for five epochs and a cosine decay schedule [46]. For 300-epoch training, we align all the hyperparameters with the official CutMix [68] for fair comparisons. Following [68], the averaged best performances are reported via three trials of experiments.

**Comparisons with State-of-the-art Regularizers.** Based on PyramidNet-200, $\tilde{\alpha}$=240 [23], Table 1 shows the comparisons against the state-of-the-art data augmentation and regularization approaches under 300 epochs. The proposed RM achieves a 2.35% Top-1 classification error in CIFAR-10, 1.5% better than the baseline (3.85%). It outperforms the two popular regularizers Mixup and CutMix by 0.74% and 0.53% points, respectively.

**Performance under Various Network Backbones.** The effectiveness of RM is further validated across a variety of network architectures in CIFAR-100 under the 200-epoch training setting, including ResNet [26], DenseNet [29], and PyramidNet [23]. From Table 3, we observe that RM has a consistent improvement in accuracy against the baselines (+1.5~3.5%) and other competitive counterparts (+0.3~1.1%). Notably, for DenseNet-161, RM shows an absolute gain of 3.2% points over the baseline model and outperforms the competitive CutMix by 1.1%.

**Ablation Study on Hyperparameter $\alpha$.** To reveal the impact of $\alpha$, we conduct ablation study by varying $\alpha \in \{0.1, 0.2, 0.3, 0.4, 0.5, 0.6, 0.7, 0.8, 0.9\}$ in CIFAR-10 with PyramidNet-200, $\tilde{\alpha}$=240 backbone under 300-epoch setting and CIFAR-100 with ResNet-18 [26] backbone under 200-epoch setting, respectively. As shown in Fig. 4, RM improves upon the baseline (3.85% and 21.70%) for all considered $\alpha$ values. The best performance is achieved when $\alpha = 0.5$.

**Ablation Study on Hyperparameter $\omega$.** Next, we examine what is the best value for the loss weight $\omega$. We adopt the same experimental setting on CIFAR-10 and CIFAR-100 as that on $\alpha$. By fixing $\alpha$ to 0.5, we change $\omega \in \{0, 0.01, 0.05, 0.08, 0.1, 0.2, 0.3, 0.4, 0.5, 1\}$ and report their performances in average value and standard deviation over 3 runs on each. Notably, when $\omega$=0, the RM model is optimized without the consistency loss, but only leverages the historical input-label pairs. Fig. 5 shows that the best performance is achieved when $\omega = 0.1$, while the performance is not sensitive (i.e., around 0.3%) in CIFAR-10 to a wide range of $\omega \in [0.1, 1]$.

**Ablation Study on RM Components.** To illustrate the performance improvement brought by each RM component individually, we first modify CutMix with the same resize strategy of RM, instead of the original cut operation. Then we add the historical mechanism and consistency loss, respectively. It is observed in Table 2 that the resize strategy can have a slightly positive effect (i.e., +0.23% accuracy) on CutMix for it is necessary to successfully formulate the correct historical (recursive) paradigm (Fig. 3). Further, the historical mechanism and consistency loss individually improve by +0.55% points and +0.16% points, respectively. Above all, RM achieves a total of 2.02% gain over the baseline and 0.94% over CutMix.

| Detector | Pretrain Backbone | AP | AP$_{50}$ | AP$_{75}$ |
|---|---|---|---|---|
| ATSS [71] | ResNet-50 [26] | 39.4 | 57.6 | 42.8 |
| | + CutMix [68] | 40.1 | 58.4 | 43.4 |
| | + RM (ours) | **41.5** | **59.9** | **45.1** |
| | PVTv2-B1 [61] | 39.3 | 57.2 | 42.5 |
| | + CutMix [68] | 41.8 | 60.3 | 45.5 |
| | + RM (ours) | **42.3** | **61.0** | **45.6** |
| GFL [42] | ResNet-50 [26] | 40.2 | 58.4 | 43.3 |
| | + CutMix [68] | 41.3 | 59.5 | 44.6 |
| | + RM (ours) | **41.9** | **60.2** | **45.6** |
| | PVTv2-B1 [61] | 40.2 | 58.1 | 43.2 |
| | + CutMix [68] | 42.1 | 60.7 | 45.5 |
| | + RM (ours) | **43.0** | **61.6** | **46.5** |

Table 7: Object detection fine-tuned on COCO with the 1× schedule by ATSS [71] and GFL [42].

| Detector | Pretrain Backbone | AP$^{box}$ | AP$^{box}_{50}$ | AP$^{box}_{75}$ | AP$^{mask}$ | AP$^{mask}_{50}$ | AP$^{mask}_{75}$ |
|---|---|---|---|---|---|---|---|
| Mask R-CNN [25] | ResNet-50 [26] | 38.2 | 58.8 | 41.4 | 34.7 | 55.7 | 37.2 |
| | + CutMix [68] | 38.5 | 58.9 | 42.2 | 34.8 | 56.0 | 37.4 |
| | + RM (ours) | **39.6** | **60.4** | **43.1** | **35.8** | **57.3** | **38.2** |
| | PVTv2-B1 [61] | 38.5 | 60.7 | 41.5 | 36.1 | 57.7 | 38.4 |
| | + CutMix [68] | 40.6 | 63.1 | 44.1 | 37.6 | 59.9 | 40.1 |
| | + RM (ours) | **41.2** | **63.5** | **44.4** | **38.1** | **60.5** | **40.9** |
| HTC [6] | ResNet-50 [26] | 41.9 | 60.5 | 45.5 | 37.1 | 57.8 | 40.1 |
| | + CutMix [68] | 42.2 | 60.7 | 46.0 | 37.4 | 58.2 | 40.4 |
| | + RM (ours) | **42.8** | **61.4** | **46.5** | **37.7** | **58.5** | **40.8** |
| | PVTv2-B1 [61] | 43.0 | 62.9 | 46.4 | 39.2 | 60.5 | 42.1 |
| | + CutMix [68] | 45.2 | 65.0 | 49.2 | 40.7 | 62.4 | 44.2 |
| | + RM (ours) | **45.8** | **65.6** | **49.8** | **41.0** | **63.0** | **44.6** |

Table 8: Object detection and instance segmentation fine-tuned on COCO with the 1× schedule by Mask R-CNN [25] and HTC [6].

**Ablation Study on Linear Classifiers.** As illustrated in Sec. 3, the historical predictions and the aligned current predictions are derived through separate linear classifiers $\mathcal{H}$ and $\mathcal{H}'$. We conducted comparisons on whether to share the parameters of the two linear layers. The training strategy on CIFAR datasets follows the above. Table 4a shows that unshared weights bring slightly better performance for it enhances model diversity which benefits the contrastive learning.

**Ablation Study on Interpolation Mode.** As depicted in Table 4b, for the resizing operation on historical inputs, we compare two common down-sampling algorithms, i.e., interpolation methods, "nearest" and "bilinear". As a result, "nearest" performs slightly better, so we fix the interpolation method to it by default.

## 4.2 ImageNet Classification

**Dataset.** The ImageNet 2012 dataset [14] contains 1.28 million training images and 50K validation images from 1K classes. Networks are trained on the training set and the Top-1/-5 errors are reported on the validation set.

**Setup.** For ImageNet, we also conduct experiments with sufficient training settings (300 epochs) under two major backbone designs, i.e., CNNs [29, 26] and Transformers [61]. For training CNN backbone networks, we employ the augmentations described in CutMix [68] to the input images for a fair comparison. All networks are trained using SGD with a momentum of 0.9, a weight decay of $1 \times 10^{-4}$, and 8 GPUs with a mini-batch size of 64 on each to optimize models. The initial learning rate is 0.2 with a linear warmup [26] for five epochs and is then decayed following a cosine schedule [46]. To optimize Transformer backbone networks, we use AdamW [47] with a learning rate of $5 \times 10^{-4}$, a momentum of 0.9, a weight decay of $5 \times 10^{-2}$, and 8 GPUs with a mini-batch size of 64 on each. We follow PVT [62] and apply random resizing/cropping of 224×224 pixels, random horizontal flipping [55], label-smoothing regularization [56], and random erasing [73] as the standard data augmentations. The hyperparameters for RM on ImageNet are set to $\alpha$=0.5, $\omega$=0.5.

**Comparison with State-of-the-art Regularizers.** Based on ResNet-50 [26], we compare RM against a series of popular regularizers in Table 6 under the 300-epoch training setting, where RM significantly improves the baseline by absolute 2.88% points in Top-1 accuracy and outperforms the strong CutMix and PuzzleMix by 0.6% and 0.44%, respectively.

**Performance under Various Network Backbones.** Since Transformer models [16, 45, 62, 61] have made significant progress in image classification due to the self-attention mechanism, we also demonstrate the effectiveness of RM on the popular Transformer network architecture PVTv2-B1 [61] in ImageNet under the 300-epoch training setting in Table 5, where it remains the superiority.

## 4.3 Transfer Learning

Benefiting from the explicit multi-scale/-space property and spatial semantic learning of the proposed RecursiveMix (RM), we suspect that the pretrained model with RM can transfer well to the downstream tasks, e.g., object detection, instance segmentation, and semantic segmentation, where multi-scale/-space semantic information plays an important role for identification.

**Object Detection and Instance Segmentation.** We conduct experiments using the one-stage object detector ATSS [71], GFL [42, 41, 40], and two-stage detector Mask R-CNN [25], HTC [6]

| Segmentor | Pretrain Backbone | mIoU | mAcc | aAcc |
|---|---|---|---|---|
| | ResNet-50 [26] | 40.90 | 51.11 | 79.52 |
| | + CutMix [68] | 40.96 | 51.16 | 79.93 |
| PSPNet [72] | + RM (ours) | **41.73** | **52.47** | **80.01** |
| | PVTv2-B1 [61] | 36.48 | 46.26 | 76.79 |
| | + CutMix [68] | 37.99 | 48.70 | 77.50 |
| | + RM (ours) | **38.67** | **49.40** | **77.93** |
| | ResNet-50 [26] | 40.40 | 51.00 | 79.54 |
| | + CutMix [68] | 41.24 | 51.79 | 79.69 |
| UperNet [67] | + RM (ours) | **42.30** | **52.61** | **80.14** |
| | PVTv2-B1 [61] | 39.94 | 50.75 | 79.02 |
| | + CutMix [68] | 41.73 | 52.99 | 80.02 |
| | + RM (ours) | **43.26** | **54.21** | **80.36** |

Table 9: Semantic segmentation fine-tuned on ADE20K [75] for 80k iterations by PSPNet [72] and UperNet [67].

| Detector | CL | AP | AP$_{50}$ | AP$_{75}$ |
|---|---|---|---|---|
| ATSS [71] | | 41.1 | 59.4 | 44.5 |
| | ✓ | **41.5** | **59.9** | **45.1** |
| GFL [42] | | 41.4 | 59.4 | 44.9 |
| | ✓ | **41.9** | **60.2** | **45.6** |

(a) Comparisons on object detection.

| Segmentor | CL | mIoU | mAcc | aAcc |
|---|---|---|---|---|
| PSPNet [72] | | 41.09 | 51.72 | 79.99 |
| | ✓ | **41.73** | **52.47** | **80.01** |
| UperNet [67] | | 41.88 | **52.79** | 79.94 |
| | ✓ | **42.30** | 52.61 | **80.14** |

(b) Comparisons on semantic segmentation.

Table 10: Comparisons on transferring downstream tasks w/ and w/o consistency loss (CL).

| ResNet-50 (300 epochs) | Memory | Flops | #P (train) | #P (deploy) | Hours | Top-1 Err (%) |
|---|---|---|---|---|---|---|
| Baseline | 5832.0 MB | 4.12 G | 25.56 M | 25.56 M | 73.0 | 23.68 |
| + Mixup [70] | 5870.0 MB | 4.12 G | 25.56 M | 25.56 M | 73.5 | 22.58 |
| + CutMix [68] | 5832.0 MB | 4.12 G | 25.56 M | 25.56 M | 73.8 | 21.40 |
| + RM (ours) | 5887.0 MB | 4.12 G | 27.61 M | 25.56 M | 73.8 | **20.80** |

Table 11: Comparisons of the training efficiency by hours, evaluated on 8 TITAN Xp GPUs. "#P" denotes the number of parameters.

in COCO [43] dataset with comparisons to the vanilla model and CutMix [68]-pretrained model based on ResNet-50 [26] and PVTv2-B1 [61]. The training protocol follows the standard 1x (12 epochs) setting as described in MMDetection [7]. In Table 7, it is observed that under ATSS, the RM-pretrained models significantly improve baseline by +2.1 AP points with ResNet-50 and +3.0 AP points with PVTv2-B1, while outperforming CutMix [68] pretrained models by +1.4 AP and 0.5 AP. Table 8 indicates the superiority of RM in instance segmentation under Mask R-CNN and HTC, where RM boosts the bbox AP and mask AP by ~2.8 and ~2.0 points, respectively.

**Semantic Segmentation.** Next, we experiment on ADE20K [75] using two popular algorithms, i.e., PSPNet [72] and UperNet [67] following the semantic segmentation code of MMSegmentation [12]. Under ResNet-50 [26] and PVTv2-B1 [61] backbone networks, we fine-tune 80k iterations on ADE20K with a batch size of 16 and AdamW [47] optimization algorithm. Table 9 shows that among various pretrained models RM outperforms the baseline and CutMix [68]. Notably, we improve the ResNet-50 by 1.9 and 1.1 mIoU compared to baseline and CutMix under UperNet. Also, there is an improvement of 3.3 mIoU over the baseline under PVTv2-B1 based on UperNet.

## 4.4 Analyses

**Class Activation Mapping Visualization.** To demonstrate the benefits of RM, we visualize the class activation map through CAM [74]. We select images that originally contain multiple categories and create CAM heatmaps for each (Fig. 6) to demonstrate the superiority of RM in spatial semantic representation. The visualization shows that RM locates the corresponding objects more accurately than Baseline and CutMix. Thanks to the recursive paradigm of RM, the algorithm can generate inputs that have multiple objects (>2), which increases the diversity of inputs in data space and enriches each instance with multi-scale and spatial-variant views. Thus, the model is more capable of recognizing complex images containing multiple categories or objects that differ greatly in size. Furthermore, the model trained under RM explicitly learns dense semantic representations and can locate the meaningful regions more accurately via the designed consistency loss. More visualization can be found in the Supplementary Material.

**RM Enables Efficient Self-distillation.** Generally, the consistency loss resembles a variant of local self-distillation by explicitly restricting two spatial representations of one instance. However, it is worth mentioning that the proposed recursive augmentation paradigm and the consistency loss are not isolated from each other. Only in the form of the "recursive mix", can we achieve self-distillation in a convenient, efficient, and memory-friendly way. Because in the recursive paradigm, the historical part in the current iteration can be supervised directly by the predictions of the former iteration as they share identical but differentially-scaled contents. It delicately fits the logic of local self-distillation, at a minimal cost of storing only the outputs of the last iteration. On the contrary, to implement local self-distillation without the recursive operation, one needs to forward one input through the networks repeatedly [64, 52, 4, 10, 24, 19, 11, 21, 57] or record the information of the whole data

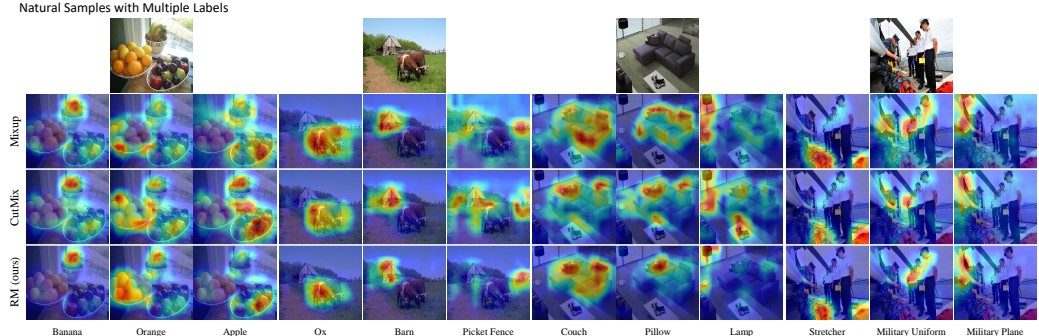

Figure 6: CAM [74] visualization comparing RM with Mixup [70] and CutMix [68] on natural samples with multiple labels.

in an entire epoch [66, 35, 38, 65], which either requires a tremendous computation cost or large memory consumption. Experimentally, RM with consistency loss enhances accuracy by 0.16% over that without in Table 2. We also verify the effectiveness of consistency loss on downstream tasks in Table 10. With the models pretrained with consistency loss, GFL and ATSS improve 0.5mAP on COCO while PSPNet and UperNet improve 0.7 mIoU on ADE20K. The results show that the introduced consistency loss enhances the performance on the downstream tasks via learning the spatial-correlative semantic representation.

**Effective Class Number.** To illustrate the diversity in the supervision signals of RM, we denote the number of classes in the regularized target whose supervision values are higher than a given threshold as effective classes. We then record the average effective number over 3 runs under 100 iterations on RM and other methods in the ImageNet training set. Note that the label smoothing threshold is roughly set to "$10^{-4}$". We suspect that if a signal value above the label smoothing threshold can be used as effective supervision, then RM can achieve a learning signal containing an average of about 4∼5 objects in a single image, which confirms the diversity of its supervision (Fig. 7).

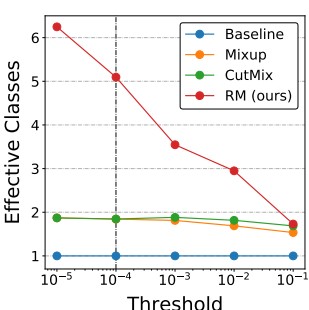

Figure 7: Comparisons of the effective classes during training under different thresholds.

**Efficiency.** We improve performance with negligible additional memory cost, as shown in Table 11. Notably, the training time and computation complexity are competitive with existing methods, which are evaluated on 8 TITAN Xp GPUs.

**Limitations.** Although our method shares the same parameters and FPS as the baseline model during employment, due to the introduction of an additional classification head and the operation of RoIAlign, we inevitably increase a small number of parameters only during the training process. In addition, it is unclear how to effectively adopt the recursive paradigm to Mixup [70] since the images would be cluttered and unidentifiable via iterative mixup operations.

## 5   Conclusion

In this paper, we propose a novel regularization method termed RecursiveMix (RM), which leverages the rarely exploited aspect of the historical information: the input-prediction-label triplets, to enhance the generalization of deep vision models. RM shares several good properties according to its historical mechanism, and it consistently improves the recognition accuracy on competitive vision benchmarks with considerably negligible additional budgets. We hope RM can serve as a simple yet effective baseline for the community.

## Acknowledgments and Disclosure of Funding

This paper is supported by the National Key R&D Plan of the Ministry of Science and Technology (Project No.2020AAA0104400), the Young Scientists Fund of the National Natural Science Foundation of China (Grant No.62206134), and the National Science Fund of China (Grant No.U1713208).

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
