# Supplementary Materials for
# "RecursiveMix: Mixed Learning with History"

**Lingfeng Yang**[1][#]**, Xiang Li**[2,1][#]**, Borui Zhao**[3]**, Renjie Song**[3]**, Jian Yang**[1][*]

[1]Nanjing University of Science and Technology, [2]Nankai University, [3]Megvii Technology

{yanglfnjust, csjyang}@njust.edu.cn, xiang.li.implus@nankai.edu.cn
zhaoborui.gm@gmail.com, songrenjie@megvii.com

## A   Broader Impacts

The proposed method is trained and will predict the results based on the statistics learned from the training datasets with potential biases that would include possible negative societal impacts. These issues warrant further research and consideration when using this technology.

## B   More Related Works of Learning with History

In this section, we review works that "Learning with History" in detail. Assume a deep model $p = f(x_0, \{W_1, ..., W_n\})$ has $n$ hierarchical layers with total $\{W_1, ..., W_n\}$ parameters, and it takes input $x_0$ to produce the output probability prediction $p$ under the current supervision $y$. The feature $x_i$ generated by each layer $i$ $(0 < i \leq n)$ constitutes the intermediate representations $\{x_1, ..., x_n\}$ at all scales. Based on the chain rule of backpropagation [26], we have two groups of gradients: $\{\nabla_{W_1}, ..., \nabla_{W_n}\}$ w.r.t. the parameters and $\{\nabla_{x_0}, ..., \nabla_{x_n}\}$ w.r.t. the input and all-level feature maps. Specifically, we use the superscript $t$ (e.g., $p^t$) to mark the moment of each variable in the training process. The superscript $t$ can be epoch-wise, iteration-wise, or customized period-wise. Given moment $t$, the historical information $H$ is then defined as a superset of all the past statistics:

$$H = \{p^j, y^j, x_0^j, \{x_1^j, ..., x_n^j\}, \{W_1^j, ..., W_n^j\},$$
$$\{\nabla_{x_0}^j, ..., \nabla_{x_n}^j\}, \{\nabla_{W_1}^j, ..., \nabla_{W_n}^j\} \mid \forall j < t\}. \tag{S1}$$

We then demonstrate a detailed review of existing representative approaches according to the above-mentioned elements.

**Prediction.** It is known that the past predictions of specific instances are from architectures with diverse parameters and distinct augmented inputs. Therefore, the prediction can carry informative and complementary probability distributions, making it potential to be a teacher that can guide the learning of the original model. Following this intuition, Kim et al. [23] adopt the prediction from only the last epoch and Laine et al. [25] successfully exploit the temporal ensembling among all the past epochs via Exponential Moving Average (EMA).

**Intermediate Feature Representation.** There are mainly two streams in this direction:

1) One stream is to directly record the instance-level feature representations, usually aiming at constructing sufficient and critical training pairs. The common practices of obtaining feature representations via computing the current mini-batch lack diversity and efficiency. To address the deficiency,

---

[*]Corresponding author. [#] Equal contributions. Lingfeng Yang, Xiang Li and Jian Yang are with Jiangsu Key Lab of Image and Video Understanding for Social Security, and Key Lab of Intelligent Perception and Systems for High-Dimensional Information of Ministry of Education, Nanjing University of Science and Technology, Nanjing, 210094, P.R. China.

36th Conference on Neural Information Processing Systems (NeurIPS 2022).

Wang et al. [38] propose a cross-batch memory (XBM) that records the instance embeddings of past iterations, allowing the algorithm to collect adequate hard negative pairs. Similarly, Wu et al. [40, 41] develop the memory bank mechanism to memorize the representations for all images to perform Neighborhood Component Analysis (NCA) [40] or Noise-Contrastive Estimation (NCE) [41] optimization, instead of exhaustively computing these embeddings every time. He et al. [16] and Chen et al. [6] implement the memory dictionary as a queue of data samples where the features are progressively replaced. Differently, Zhong et al. [53] store the feature maps of *unlabeled data* from the target domain, introducing the exemplar-invariance objective with the source domain to bridge the domain gap.

2) Another stream is to memorize feature statistics, e.g., moving mean and variance. Wen et al. [39] update the category-level feature centers and penalize the distances between the deep features and their corresponding class centers, enabling stronger intra-class compactness. In Ioffe et al. [21], the mean and variance statistics calculated and accumulated inside each Batch Normalization layer across all levels of features, are utilized during model inference. Caron et al. [2] maintain the first-order batch statistics as adding a bias term to the teacher network, successfully avoiding the training collapse.

**Model Parameter.** The historical usage of model parameters can be divided into three major groups:

1) Constructing the teachers from past models to effectively supervise the students. The popular practice is to build a teacher network by leveraging the historical student parameters in unsupervised [2, 6, 16, 11, 7, 14], supervised [45, 27, 44, 12], and semi-supervised [43, 35] learning. The resulted teacher models can provide sufficient and informative supervision for the original model to distill, significantly boosting the performance of the student models.

2) Directly exploiting ensemble results in inference from multiple past models. Huang et al. [19] and Chen et al. [3] store a set of model snapshots (or checkpoints) to perform multiple evaluations for each test sample, then ensemble the predictions as the final results with improved accuracy.

3) Building unitary ensemble architecture in inference from previous models. Different from [19, 3] where multiple networks are adopted for inference, [48, 1, 31, 13, 22] attempt to average/ensemble the historical model parameters directly and obtain a final single structure for efficient deployment.

**Gradient.** The advanced optimizers commonly rely on the statistics of historical gradients in a way of EMA for speeding up the training convergence, where momentum SGD [32] calculates the first-order statistics, Adagrad [10] utilizes the second-order one whilst Adadelta [47], Adam [24], AdamW [30] and RAdam [29] make use of both.

Different from existing practices, we explore a novel usage of historical information by leveraging the input-prediction-label triplets to improve the generalization of vision models.

## C Implementation Details

### C.1 Architectures

**Convolutional Neural Networks.** We follow the official implementation of each CNN network, including ResNet [18], DenseNet [20], and PyramidNet [15]. For CIFAR experiments, we remove the max-pooling layer and replace the first 7×7 kernel with a size 3×3. As for the ImageNet experiments, the standard implementation of all models is employed.

**Vision Transformers.** We adopt the popular PVT architecture in our experiments. The implementation follows PVTv2 [36]. Notably, we use a spatial average pooling operator instead of the linear projection in the attention blocks for its higher efficiency.

### C.2 Image Classification on CIFAR and ImageNet-1K

The CIFAR-10 and CIFAR-100 datasets consist of 60K 32x32 color images in 10 and 100 classes, respectively. ImageNet-1K [9] contains 1.3M images divided into 1K classes. Only CNN models participate in CIFAR experiments, which are trained under 200-epoch and 300-epoch settings following CutMix [46]. In terms of ImageNet-1K experiments, we refer to the training setting of CNN to CutMix [46] and Transformers to PVT/PVTv2 [37, 36]. In detail, we employ SGD for CNNs and AdamW [30] for Transformers. As for the data augmentation, besides the random

resizing/cropping of 224×224 pixels and random horizontal flipping [33] for CNNs, the label-smoothing regularization [34] and random erasing [52] are additionally adopted for Transformers.

## C.3 Object Detection on COCO

The COCO datasets contain annotations in 80 categories, with over 1.5 million object instances. We adopt one-stage detectors ATSS [50], GFL [28], and two-stage detectors Mask R-CNN [17], HTC [4] to compare the transfer performance of RM pretrained models with baseline and CutMix [46]. The experimental setting is a standard 1x (12 epochs) schedule following MMDetection [5] and the optimizers for CNN and Transformer models are SGD and AdamW, respectively.

## C.4 Semantic Segmentation on ADE20K

The ADE20K dataset contains 25K images annotated with 150 object categories. We employ PSPNet [51] and UperNet [42] following MMSegmentation [8] under the 8K-iteration training setting. Notably, we adopt AdamW [30] as the optimizer for CNN models for faster convergence and better performance (Table S1).

| Optimizer | Pretrain Backbone | mIoU | mAcc | aAcc |
|---|---|---|---|---|
| SGD | ResNet-50 [18] | 39.82 | 50.69 | 79.65 |
| | + CutMix [46] | 40.22 | 51.72 | 79.80 |
| | + RM (ours) | **41.44** | **52.91** | **79.99** |
| AdamW | ResNet-50 [18] | 40.40 | 51.00 | 79.54 |
| | + CutMix [46] | 41.24 | 51.79 | 79.69 |
| | + RM (ours) | **42.30** | **52.61** | **80.14** |

Table S1: Comparisons between SGD and AdamW on semantic segmentation with UperNet [42].

# D  Examples of Augmented Data

To give a detailed and specific view of the augmented data generated by the proposed RC method, we randomly illustrate the training samples (with $\alpha = 0.5$) in the ImageNet dataset, as shown in Fig. S1. The order is left to right, top to down. For every single instance, we observe at least 2∼3 clear different scales and positions of it along the sequence, depending on how fast it vanishes according to the sampled ratio $\lambda$. Therefore, we believe that the training can indeed benefit from the rich supervision and the enlarged data space with explicit multi-scale/-space views.

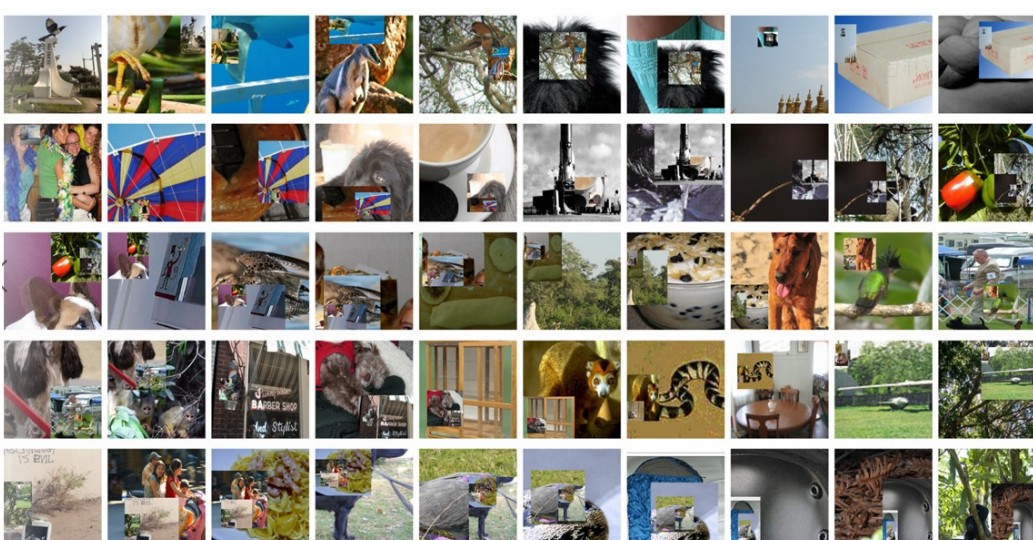

Figure S1: Examples of the augmented training data produced by Recursive CutMix in ImageNet. Read from left to right, top to bottom.

In addition, we visualize the activation map for each participating ground truth class in the mixed images generalized by RM (Fig. S2).

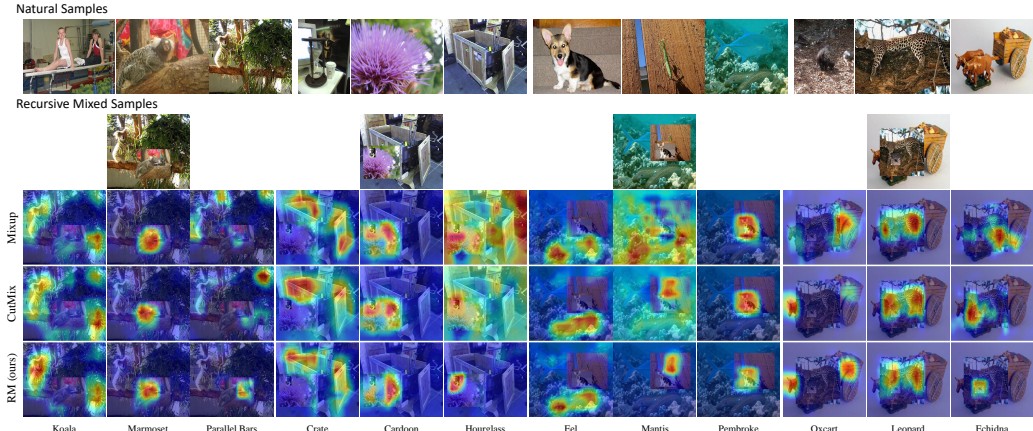

Figure S2: CAM [54] visualization comparing RM with Mixup [49] and CutMix [46] on samples created using RM.