# OpenReview forum: "RecursiveMix: Mixed Learning with History"
_NeurIPS.cc/2022/Conference — NeurIPS 2022 Accept_

### Official Review · Reviewer_oBX3 · 2022-07-10

**Rating:** 6
**Confidence:** 4
**Soundness:** 4 excellent
**Presentation:** 3 good
**Contribution:** 3 good

**Summary:**

This paper presents RecursiveMix that applies CutMix iteratively. The proposed method is efficient and effective and shows strong generalization ability for different tasks including image classification, object detection, instance and semantic segmentation.

**Questions:**

See weaknesses

**Limitations:**

See weaknesses

**Strengths And Weaknesses:**

### Strengths

1. The idea is simple and effective, which makes it easy to follow.

2. The performance it achieves is impressive. It can lead to consistent gain for four different tasks and all of them were evaluated on challenging benchmarks. Meanwhile, it only requires marginal computational cost during training and is completely cost-free during inference.

3. The experiments and ablation studies are comprehensive and extensive. The authors evaluate their methods on a wide range of tasks and datasets, which can help to attract more audiences with different backgrounds.


### Weaknesses

1. RecursiveMix changes the previous crop-based CutMix to resize-based CutMix and keeps the original label assigning strategy in CutMix, which seems counter-intuitive. In this way, the foreground area of the crop in the mixture will be always kept since it will resize the whole image and keep all content there. But the crop can cover the foreground area of the other image, which means the valid object can sometimes disappear. Intuitively, the label assignment under such circumstances should always keep the label of the cropped image as 1 since the foreground object in the crop always exists, and the label confidence for the other image in the mixture should be lower.

---

> ### Author Response · Authors · 2022-08-02
> **Response to R5**
>
> Thank you for the comments and suggestions!
>
> **Q1:** Changing crop-based CutMix to resize-based CutMix and keeping the original label assigning strategy in CutMix seems counter-intuitive.\
> **A1:** We answer these questions from the following two aspects. \
> ***1)	Why do we keep the original label assigning strategy?*** \
> Ideally, the label assigning strategy of all mixed instances should ***both consider their semantical integrity and the proportion of their composition area***. Although the resized foreground mixture of the image keeps its full semantical content, its area is reduced. Think of an extreme condition where the foreground area is resized to a small ratio like 0.01, which is almost invisible on the input. Then keeping its confidence to 1 will be inappropriate. Therefore, we keep the original label assigning strategy, which considers the actual area of each mixed instance in an image.\
> ***2)	Why do we change from crop-based to resize-based?*** \
> From a qualitative perspective, a “Resize” operation will keep the former information, and ensure consistency of the label (Fig.3). From an experimental perspective, the “Resize” strategy surpassed the “Cut” strategy by 0.23% accuracy.

---

### Official Review · Reviewer_1VFL · 2022-07-15

**Rating:** 4
**Confidence:** 5
**Soundness:** 2 fair
**Presentation:** 2 fair
**Contribution:** 2 fair

**Summary:**

The authors of this paper propose to increase the model generalizability via historical mix-up. Specifically, the compositional training batch from previous time-step is stored and ultilized in the training process at current time-step.

**Questions:**

Please refer to Sec.2.

**Limitations:**

Please refer to Sec.2.

**Strengths And Weaknesses:**

Weaknesses:

1. Is there any experimental comparison with hierarchical mixed-up in large-size batch? Specifically, the comparison with hierarchical mixed-up within single batch, rather than using the last time-step information. Since during the training process of large datasets, the number of batch in each epoch could be large, which may cause two problems:
a) Over-memorizing the information(images) appears in the early batches;
b) Unexpected semantic dependencies caused by certain information co-occurance.

2) Line38. The proposed recursive-mix improves the inner-data diversity within a batch, while decrease the intra-data diversity within an epoch by information paste.

3) Although the increase of RM's model complexity compared with others is small, is there any possible that the performance gain comes from the increased model complexity? Is there any experiment to verify this assumption?

4) In Eq(5). a) Since H and H' are with different weights, which part of the model is actually optimized for distribution alignment? The encoder ot the projector H?
b) The network is updated after processed each batch, will the KL-term affacts/prevent the optimization process? Another possible is that it will result in updating momentum appears in contrative learning methods, is there any possible that the performance gain actually comes from the momentum rather than the mix-up process?
c) The information loss is ignored during the mix-up process. Even an instance can still be observed at current stage, it is hard to ensure that the information loss is aligned with the scale loss. For example, without the tiny boarders, a billboard will be same with the wall, thus its key information may disappear within few time-steps.

---

> ### Author Response · Authors · 2022-08-02
> **Response to R4**
>
> Thank you for the comments and suggestions!
>
> **Q1:** Concern about the large batch training of RM.\
> **A1:** ***First of all***, we would like to clarify the concerns over the large batch training of RM by answering the following questions.\
> ***1)	Are there differences between a small batch and a large batch for RM?*** \
> ***RM operates the same under different batch sizes***. Above all, all the recursively resized operations are done between each data pair of the same position within a batch. Therefore, the chance one instance is memorized by the model only depends on the random resized ratio, rather than the batch size.\
> ***2)	Will RM over-memorize the information (images) that appears in the early batches?*** \
> RM will not over-memorize the historical information. ***Firstly***, the area and assigned label weight of a historical instance will reduce every iteration and finally become invisible on the input, so it will not be over-memorized by the model (See the visualization of the mixed images in the Supplementary Material). ***Then***, As discussed in 1), a larger batch will not increase the memory process of the information in the early batches.\
> ***3)	Will there be unexpected semantic dependencies caused by certain information co-occurrence?*** \
> The possibility is low. On one aspect, in visual tasks, the possibility of information co-occurrence is quite low due to the rich variety of classes. From our visualization in the Supplementary Material, we hardly observe this certain situation. On the other aspects, information co-occurrence is a potentially common problem that may occur in existing pixel-based mixed augmentations and is not yet ideally solved.
>
> ***Secondly***, we conduct experimental comparisons on large batch training with the hierarchical mix-up under 200 epoch settings based on ResNet-18 on CIFAR-100. Notably, we choose a batch of 1024 and 2048.
>
> | ResNet-18| batch=1024 | batch=2048 |
> | :--- | :--------: | :--------: |
> | Baseline|   0.7721   |   0.7467   |
> | Mixup    |   0.7755   |   0.7575   |
> | CutMix|   0.7815   |   0.7652   |
> | Manifold Mixup|   0.5965   |   0.5891   |
> | RM|   0.7973   |   0.7690   |
>
> Manifold Mixup is originally trained with ***a batch size of 100 for 2000 epochs*** on CIFAR. However, under the common 200 epochs setting with a large batch, severe performance degradation is observed on Manifold Mixup. Meanwhile, from the above table, RM still maintains the best performance.
>
> **Q2:** Will RM decrease the intra-data diversity within an epoch by information paste?\
> **A2:** ***Pasting information from the last batch will not decrease the intra-data diversity***. On the contrary, RM generates inputs with more diverse samples and benefits the training. To demonstrate this conclusion, we conduct ablation studies on Mixup and Cutmix. Originally, Mixup and Cutmix mixture two random images within the current batch. Then, we make one modification on Mixup and Cutmix that two images are mixed between the last batch and current batch while maintaining their mixing property. The modified version pastes information from the last batch, just like RM. We follow the official 200 epochs training setting under ResNet-18 on CIFAR100.
>
> | ResNet-18 | Top-1 Acc |
> | :------- | :-------: |
> | Baseline  |  0.7830   |
> |   Mixup   |  0.7901   |
> |  Mixup*   |  0.7932   |
> |  CutMix   |  0.8039   |
> |  CutMix*  |  0.8041   |
> |    RM     |  0.8136   |
>
> “``*``” denotes the modified mix-methods. The table shows that pasting from the current or last batch will not affect the performance and even improves a little. Therefore, ***pasting historical information does not harm the intra-data diversity***.
>
> **Q3:** Is there any possibility that the performance gain comes from the increased model complexity? \
> **A3:** Based on our paper, the performance gain does not come from the increased model complexity for two reasons. (1) During inference, the employed model is the same as the baseline, there is no “increased model complexity” (Table 11). (2) In Table 4 (a), we conduct ablation studies that even keep the same model parameters during training by a shared classification head, which shows that RM can still surpass CutMix and Mixup.
>
> **Q4:** Which part of the model is optimized for distribution alignment? \
> **A4:** The entire model parameters, except the normal classifier head, are optimized for distribution alignment.
>
> **Q5:** Will the KL-term affect/prevent the optimization process? Will it result in updating momentum appearing in contrastive learning methods? \
> **A5:** The KL-divergence will promote the optimization process. Intuitively, it minimizes the distance between two features that are derived by networks with different parameters from an identical instance. Therefore, the model gains better spatial semantical representation ability.

---

> > ### Author Response · Authors · 2022-08-02
> > **Response to R4 (Part 2)**
> >
> > **Q6:** Does the performance gain comes from the momentum rather than the mix-up process?\
> > **A6:**
> > The proposed recursive augmentation paradigm and the consistency loss (i.e., self-distillation) are not isolated from each other. ***Only in the form of the “recursive mix”, can we achieve self-distillation in a convenient, efficient, and memory-friendly way***. Because in the recursive paradigm, the historical part in the current iteration can be supervised directly by the predictions of the former iteration as they share identical but differentially-scaled contents. It delicately fits the logic of local self-distillation, at a minimal cost of storing only the outputs of the last iteration. On the contrary, to implement local self-distillation without the recursive operation, one needs to additionally forward the inputs through the networks or record the information of the whole data in an entire epoch, which either requires a tremendous computation cost or large memory consumption.
> >
> > We’d like to clarify that ***our improvements are gained both from the recursive mix-up process and the semantically aligned optimization (momentum)***, which shows the superiority of RM jointly. Table 2 shows the detailed improvements of each component in RM. ***Firstly***, The resize strategy and historical mechanism have a positive effect of +0.23% and +0.55% over CutMix, respectively. ***A total of 0.78% accuracy is gained from the mix-up process***. ***Then, adding the KL-divergence further improves +0.16% points, denoting the gain from the optimization process/momentum***.
> >
> > **Q7:** Even if an instance can still be observed at the current stage, it is hard to ensure that the information loss is aligned with the scale loss.\
> > **A7:** ***Firstly***, compared to the CutMix, we replace the “Cut” operation with “Resize”, which can correctly preserve the consistency and alleviate the information loss (Fig.3). ***Secondly***, the label weight is proportional to the area of instances (scale), ensuring that a visually larger instance gets a higher label weight.

---

### Official Review · Reviewer_kceg · 2022-07-16

**Rating:** 5
**Confidence:** 4
**Soundness:** 3 good
**Presentation:** 2 fair
**Contribution:** 3 good

**Summary:**

This paper proposes a novel data augmentation strategy (RecursiveMix). The proposed augmentation recursively resizes the historical input and then fills it into the current batch. Experiments demonstrate that the proposed RecursiveMix consistently outperforms the popular augmentations such as Mixup and CutMix.

**Questions:**

Please address the above weaknesses.

**Limitations:**

The authors have discussed the limitations. No potential negative societal impact.

**Strengths And Weaknesses:**

Strengths:
1. The proposed augmentation is novel and interesting.
2. Abundant experiment results are provided. According to the experiment results, RecursiveMix can consistently improve the performance.

Weaknesses:
1. The discussion subsection 3.2 is not informative. Most information has been included in the introduction.

2. The paper also lacks an experiment in which several basic augmentations, e.g. cropping, scaling and flipping, have been applied and then the authors further add and compare RecursiveMix to advanced augmentations, e.g. mixup and cutmix. In other words, the authors should consider the joint usage of multiple augmentations.

3. It will be better to give the standard deviation of multiple experiments.

4. The proposed method involves two hyper-parameter \alpha and \omega. The ablation study shows that the performance is sensitive to the hyper-parameters considering that the improvements over Mixup and CutMix are usually less than 1%.

Small problems:
1. The font in figures can be larger for reading easily.
2. The interval between bars in Figure 1 can be uniform.
3. Syntax error in Line 129 “Because …”

---

> ### Author Response · Authors · 2022-08-02
> **Response to R3**
>
> Thank you for the comments and suggestions!
>
> **Q1:** The discussion subsection 3.2 is not informative. Most information has been included in the introduction.\
> **A1:** Thanks for your suggestion! We will rearrange the content in the revised version.
>
> **Q2:** The paper lacks the experiment in which several basic augmentations, e.g., cropping, scaling, and flipping, have been applied and then further add and compare RM to advanced augmentations.\
> **A2:** By default, in all experiments, ***we have already applied random cropping, scaling, and flipping***. Also, more augmentations such as label-smoothing, rand-aug,.etc. are employed. In Sec.4.1 – Setup, Sec.4.2 – Setup, and the Supplementary Material, all augmentations are elaborated and precisely attached to corresponding works.
>
> **Q3:** It will be better to give the standard deviation of multiple experiments.\
> **A3:** We will add the standard deviation in the revised version. Part of the standard deviation is already depicted in the ablation study of Sec.4.1 (Fig.4 & Fig.5).
>
> **Q4:** The performance is sensitive to the hyperparameters considering that the improvements over Mixup and CutMix are usually less than 1%.\
> **A4:** We would like to discuss the sensitivity of the hyperparameter in turn.\
> ***1)	The sensitivity of the resizing ratio α.*** \
> Considering that α decides the resize ratio, a small α degenerates RM to the baseline, therefore it is truly normal that the accuracy drops a large margin when α switches from 0.5 to 0. It instead shows RM can significantly improve the baseline. However, when α is 0.5~0.7, the performance is fairly stable (±0.2).\
> ***2)	The sensitivity of the consistency loss weight ω.*** \
> Firstly, the ω is stable on CIFAR-10 (±0.2). Then for CIFAR-100, a less-optimum ω surpasses the CutMix by ~0.5%, and the best ω surpasses by 1%, which are both promising improvements over the CutMix.
>
> **Q5:** Other minor issues.\
> **A5:** Thanks! We will carefully address them in the revised version.

---

> > ### Comment · Reviewer_kceg · 2022-08-05
> > **Thanks for the response.**
> >
> > Thanks for the response. The response answered my question about the mixed use of augmentations. However, I still think the two hyper-parameters weaken this paper. I would think this is borderline work.

---

### Official Review · Reviewer_HH8z · 2022-07-17

**Rating:** 7
**Confidence:** 4
**Soundness:** 3 good
**Presentation:** 3 good
**Contribution:** 3 good

**Summary:**

This paper introduces a recursive data mixing augmentation called RecursiveMix. It pastes a historical image patch onto the current training sample to promote the data diversity for image mixing. Furthermore, a consistency loss is introduced to force the local pasted patch prediction is invariant to the background. Extensive evaluation on image classification, object detection, and semantic segmentation shows that RM largely improves performance.

**Questions:**

the proposed method needs an additional task head, which inevitably adds on the training computation. However,  the authors only compare the cost of the classification task, with a linear head. In contrast, the task head in detection and segmentation is extremely large compared to the backbone. Could the author please compare the computation cost overhead on detection and segmentation?

**Limitations:**

The limitation is discussed in Line 313-319.

**Strengths And Weaknesses:**

# Strength
1. The method is well-motivated and finely designed. The two key components are tightly coupled: recursive data mixing promotes image diversity, while local patch consistency calls for representation invariance regardless of background.
2. The experiments are truly comprehensive. The author not only testifies the performance on a wide range of mainstream visual tasks but also provides an extensive ablation study on each component of the method.
3. The proposed method shows compelling improvements compared with existing methods.
4. The paper is clearly written and easy to follow.

# Weakness
1. The current method seems not to provide striking new insights. The current RM design is mainly a improved version of exiting data mixing pipeline, by considering the historical data information. I do not mean the derived approach is "not novel". I just want to see more fundamental insights or rigorous discussion on "why the current design is reasonable?" and "How does it improve the performance?". Good performance itself just indicates the results, but tell nothing about the reason.
2. No ablation on $\lambda$. In lines 127-131, the author claims that "$\lambda$ denotes the proportion of the historical images, which is not suggested to be quite large". However, no ablation is provided on resizing ratio $\lambda$. I strongly suggest providing the results.

---

> ### Author Response · Authors · 2022-08-02
> **Response to R2**
>
> Thank you for the comments and suggestions!
>
> **Q1:** More fundamental insights or rigorous discussion of RM.\
> **A1:** About the insights and discussion on the classification pipeline of RM, we answer the following questions in turn.\
> ***1) Why is the current design reasonable?*** \
> Notably, the proposed RM design is not only an improved version of a data mixing pipeline. ***Firstly***, we design a smart recursive paradigm, that utilizes both the historical ***input-prediction-label triplets***, rather than merely mixing upon the image data. ***Secondly***, multiple-mixed data will provide more diverse training samples, which is already demonstrated by dozens of mixed augmentation strategies. ***Thirdly***, RM creates multi-scale instances during training, which is proved workable by various visual tasks like object detection. ***Finally***, RM managed to learn the spatial semantics via the consistency loss because it minimizes the distance within each semantic pair. However, the existing mixed augments only provide image-level annotations, regardless of the spatial distribution of the mixed semantics. \
> ***2) How does it improve the performance?*** \
> As shown in Table 2, the performance is improved by three components. ***Firstly***, by replacing the “Cut” operation with “Resize” in CutMix, we improve +0.23% over CutMix. Because a “Resize” operation will keep the former information, and ensure consistency of the label (Fig.3). ***Secondly***, the historical mixed paradigm improves further improves +0.55% accuracy. Because RM provides more variant instances and multi-scale semantics. ***Finally***, adding the consistency loss additionally improves +0.16%, for it enhances the spatial semantical representation ability. In total, RM surpasses the baseline and CutMix by 2.02% and 0.94%, respectively.
> More discussions can be found in Sec.3.2 and Fig.2.
>
> **Q2:** No ablation is provided on the resizing ratio λ.\
> **A2:** ***λ is not a hyperparameter***. It is randomly sampled from the uniform distribution U[0,α] at each iteration, and we have conducted an ablation study on α (Fig.4).
>
> **Q3:** Could the author please compare the computation cost overhead on detection and segmentation?\
> **A3:** The computation cost for detection and segmentation of RM is exactly ***the same as the baseline***. Because the additional auxiliary head is only adopted during the classification training process. Once we get the pretrained models, we drop the auxiliary head and only employ models with the original head for downstream tasks. Therefore, the computation cost is the same as other methods.

---

> > ### Comment · Reviewer_HH8z · 2022-08-06
> > **Thanks for the Reponse And One More Question**
> >
> > Thanks for the response. It partially addresses my concerns.
> >
> > One more question: Did you try to **replace the KL divergence with cross-entropy on the one-hot label**? Because for now, you are actually doing a variant of **local self-distillation**. I am not sure it is the new augmentation makes the improvement or if it is just the self-distillation works. Thank you again for your hard work.

---

> > > ### Author Response · Authors · 2022-08-08
> > > **Response to Reviewer HH8z**
> > >
> > > **1) Did you try to replace the KL divergence with cross-entropy on the one-hot label?** \
> > > In fact, replacing the KL divergence with cross-entropy on the one-hot label may not be optimal for supervising the RoI Aligned region. Because this area contains multiple classes accumulated from the recursive operation on images, rather than a one-hot label. Therefore, a multi-label target is necessary for optimization. Further, in our case, optimizing the KL divergence is identical to cross-entropy theoretically, since the multi-label target of the historical input is detached from calculating gradients. Specifically, by denoting the prediction at the current iteration as $p_i$, we prove that KL loss is equivalent to CE plus a Const, where the Const can be ignored during optimization.
> > > \begin{aligned}
> > > &\boldsymbol{C E}=-\sum_{i} y_{i} \cdot \log \left(p_{i}\right)
> > > \end{aligned}
> > > \begin{aligned}
> > > &\boldsymbol{K L}=-\sum_{i} y_{i} \cdot \log \left(\frac{p_{i}}{y_{i}}\right)=-\sum_{i} y_{i} \cdot \log \left(p_{i}\right)+\sum_{i} y_{i} \cdot \log \left(y_{i}\right)=\boldsymbol{C E}+\boldsymbol{C O N S T}
> > > \end{aligned}
> > >
> > >
> > > **2) Does the new augmentation make the improvement or if it is just the self-distillation works？** \
> > > The proposed recursive augmentation paradigm and the consistency loss (i.e., self-distillation) are not isolated from each other. ***Only in the form of the “recursive mix”, can we achieve self-distillation in a convenient, efficient, and memory-friendly way***. Because in the recursive paradigm, the historical part in the current iteration can be supervised directly by the predictions of the former iteration as they share identical but differentially-scaled contents. It delicately fits the logic of local self-distillation, at a minimal cost of storing only the outputs of the last iteration. On the contrary, to implement local self-distillation without the recursive operation, one needs to additionally forward the inputs through the networks or record the information of the whole data in an entire epoch, which either requires a tremendous computation cost or large memory consumption.
> > >
> > > In fact, they both contribute to the improvement, where the new augmentation accounts for the majority. Table 2 shows the detailed improvements of each component in RM, that the ***recursive augmentation*** and ***KL-divergence*** improve by ***0.78%*** and ***0.16%*** over CutMix, respectively.

---

> > > > ### Comment · Reviewer_HH8z · 2022-08-08
> > > > **Thanks for the feedback**
> > > >
> > > > Thanks for the clarification. I understand the "recursive mix" allows the efficient form of self-distillation. But this point is never mentioned in the paper until I raise it. I suggest the authors provide a brief discussion in the paper.
> > > >
> > > > Best Wishes

---

> > > > > ### Comment · Reviewer_HH8z · 2022-08-08
> > > > > **Increase the rating**
> > > > >
> > > > > After careful consideration, I decided to increase my rating by one score.
> > > > > I hope that our discussion could be carefully resolved in the final revision.
> > > > >
> > > > > Best

---

### Official Review · Reviewer_3KBU · 2022-07-20

**Rating:** 7
**Confidence:** 4
**Soundness:** 3 good
**Presentation:** 3 good
**Contribution:** 3 good

**Summary:**

This paper proposes an interesting idea on data augmentation via mixture of samples. Unlike typical ways of mix-based data augmentation methods, this work performs iterative mixtures, which is able to reuse the augmented mixed samples and thereby generates more diverse samples. The idea is simple yet effective, which is validated by extensive experiments.

**Questions:**

1. Is such recursive mixture operation performed the same times of iterations of the whole optimization? Or is there an upper-bound on the recursive times? Intuitively, more recursive times leads to richer and more diverse semantics in the augmented samples, however, incorporating too many  classes of semantics may degenerate the performance. Maybe it is better to conduct experiments to investigate the effect of recursive times of mixture operations.

2. It is claimed that the introduced consistency loss can help the model to learn the spatial-correlative semantic representation, which potentially benefit the downstream tasks like object detection or semantic segmentation. Thus, it is better to conduct such ablation study to verify the effect of the consistency loss on these two tasks.

**Limitations:**

Refer to the questions posed above.

**Strengths And Weaknesses:**

Strengths:
1. The idea of recursive mixtures of samples is novel.
2. Extensive experiments demonstrates the effectiveness of the method.
3. The paper is written well and easy to follow.

---

> ### Author Response · Authors · 2022-08-02
> **Response to R1**
>
> Thank you for the comments and suggestions!
>
> **Q1:** Is there an upper bound on the recursive times? Will more recursive times incorporate too many classes of semantics and degenerate the performance? \
> **A1:** No, there are no limitations on the recursive time, i.e., it performs the same times of iterations as the whole optimization. ***Generally***, this operation won’t be incorporating too many classes, because the former instance will visually disappear from the input after more than about five iterations since its area and activation signal reduce with each iteration. ***Specifically***, as is illustrated in Sec.4.4 Analyses - Effective Class Number: at most, there are only an average of 5 objects in a single image that is semantically supervised (Fig.8). Therefore, the upper bound of classes at a time is restricted naturally, and the performance will benefit from the richer and more diverse semantics.
>
> **Q2:** Missing the ablation study to verify the effect of the consistency loss on downstream tasks like object detection or semantic segmentation. \
> **A2:** We have conducted this ablation study in Table 10 (page 8). With the consistency loss, GFL and ATSS improve ~0.5mAP on COCO while PSPNet and UperNet improve ~0.7 mIoU on ADE20K. The results show that ***the consistency loss enhances the performance of the downstream tasks***.

---

> > ### Comment · Reviewer_3KBU · 2022-08-10
> > **My concerns are addressed**
> >
> > Thanks for the response, which has addressed my concerns. Thus, I won't change my rating.

---

### Meta-Review · Area_Chair_KAy4 · 2022-08-25

**Recommendation:** Accept
**Confidence:** Certain

**Metareview:**

The manuscript has been reviewed by five reviewers with ratings of 4,5,6,7,7. The reviewers are in general happy with the contributions, novelty, experimental validation, and mostly recommended acceptance. The AC agrees with the majority vote and would like to recommend acceptance. Congratulations!



**Award:**

No

---

### Decision · Program_Chairs · 2022-09-14

Accept